# Provenance Indication of Rare Earth Elements in Lake Particulates from Environmentally Sensitive Regions

Pu Zhang [1,*,†], Zhe Zhang [2,†], Lihua Liang [3], Lei Li [1], Chenyang Cao [4] and R. Lawrence Edwards [5]

1   International Center for Planetary Science, College of Earth Sciences, Chengdu University of Technology, Chengdu 610059, China; 13969051202@163.com
2   College of Environmental Science and Engineering, Nankai University, Tianjin 300350, China; zzheedu@163.com
3   College of Urban and Environmental Sciences, Northwest University, Xi'an 710127, China; lianglh@nwu.edu.cn
4   School of Ocean and Earth Science, Tongji University, Shanghai 200092, China; caochenyang@stumail.nwu.edu.cn
5   Department of Earth and Environmental Sciences, University of Minnesota, Minneapolis, MN 55455, USA; edwar001@umn.edu
*   Correspondence: zhangpu035@cdut.edu.cn
†   These authors contributed equally to this work.

**Abstract:** The provenance of lake particulate matter in environmentally sensitive areas is crucial to understanding regional environmental and climatic changes. This study investigated two regions in the Northeast Tibetan Plateau, China: Region I (Keluke, Tuosu, and Gahai Lakes) and Region II (Qinghai Lake and nearby rivers). The results showed that: (1) The two regions have greater differences in the enrichment of rare earth elements (REEs) and heterogeneity in spatial distribution, both of which are characterized by relative enrichment of LREE and depletion of HREE, but to different degrees; (2) the source and formation of particulate matter in two regions are consistent. Particulate matter in Region I (Keluke and Tuosu Lakes) predominantly originates from granite rocks, which undergo weathering and transportation through rivers. Region II (Qinghai Lake and nearby rivers) particulate matter is affected by chemical weathering and partial recycling of detrital material. Diagenesis had a minimal impact on the particulate REEs. (3) This study primarily provides a preliminary understanding of REEs in lake particles, assessing particle changes during the water-to-sediment process and their provenance indication. Future studies will incorporate the solid fugacity (solid speciation) of REEs in particles, contributing to a comprehensive understanding of rare earth element geochemical processes. This study provides valuable insights into REEs distribution, source, and geochemical behavior in the Tibetan Plateau, underscoring the importance of REEs in understanding provenance processes, and is indicative of provenance studies in other climate change-sensitive regions of the world.

**Keywords:** rare earth elements (REEs); provenance; particulate matter; qinghai lake; keluke lake; tuosu lake





## 1. Introduction

The lake, as an important component of terrestrial aquatic ecosystems, contains rich information about geology, the environment, and human activities. The study area is located in the Northeast Tibetan Plateau, where lakes are of diverse types, including saline and freshwater lakes, and are sensitive to climate change (Figure 1a). For instance, fluctuations in temperature can exert a significant impact on the ecological balance of these lakes [1]. Furthermore, processes such as snowmelt, glacial runoff, and precipitation can result in water level rise and water quality variations in these lakes [2–4]. Consequently, the distinctive ecosystems of plateau lakes contain abundant information on geological records and environmental changes.

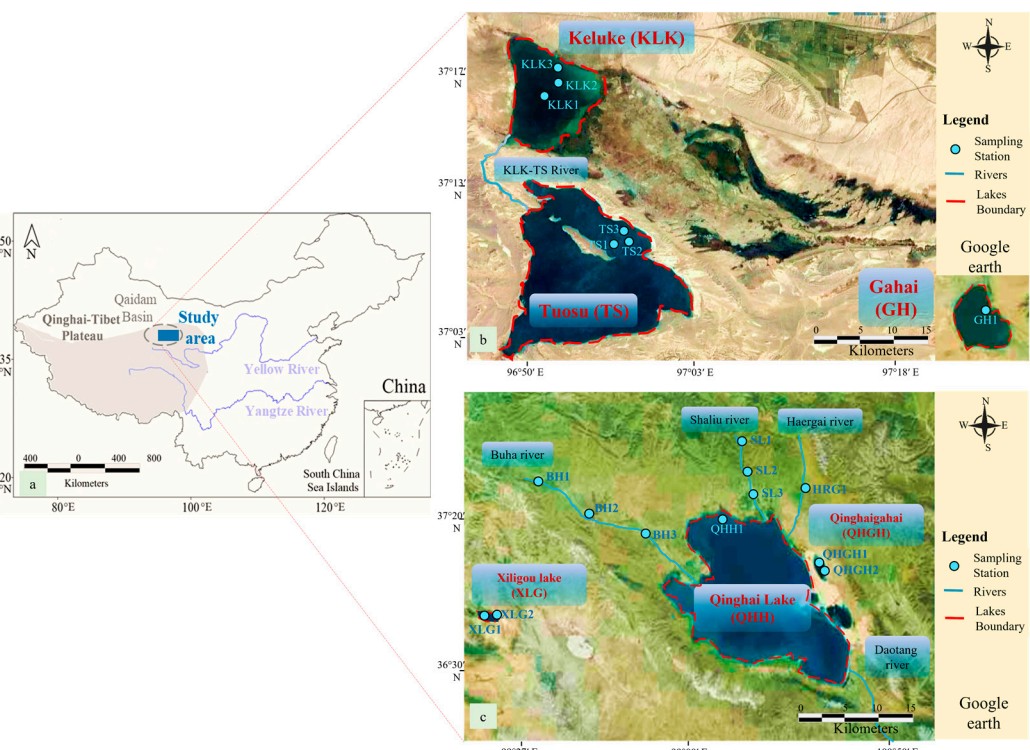

**Figure 1.** Location of the study area. (**a**) Schematic map of the Northeast Tibetan Plateau in China. (**b**) The detailed sampling stations of Keluke Lake, Tuosu Lake, and Gahai. (**c**) The detailed sampling stations of Qinghai Lake, Qinghaigahai, Xiligou Lake, Shaliu River, Haergai River, and Buha River.

Currently, methodologies employed for investigating provenance encompass mineralogy [5], geochemistry [6], radioactive decay tracing [7], isotope tracing [8], etc. Among them, rare earth elements (REEs) are extensively applied in provenance studies. REEs represent a specialized group within the geochemical periodic table, consisting of 15 elements spanning from lanthanum (La) to lutetium (Lu) and yttrium (Y). They are typically divided into two categories: light rare earths (LREE, La, Ce, Pr, Nd, Sm, and Eu) and heavy rare earths (HREE, Gd, Tb, Dy, Ho, Er, Tm, Yb, and Lu). Studies have demonstrated that these elements vary continuously with increasing atomic number and are more traceable than single-trace element traces [9]. Since they are not easily transported in natural environments and are stably distributed and systematic, they provide key information for analyzing sediment genesis, origin, and formation conditions, especially in provenance tracing.

Sediment alteration and transport dynamics within the benthic boundary layer of lakes were traced early by REEs [10]. Subsequently, an increasing number of studies have been conducted. Wang et al. [11] investigated the distribution and sources of REEs in the surface and core sediments of Dongting Lake, emphasizing the significance of REEs in tracking sediment transport and contaminant sources within aquatic environments. Meanwhile, Sojka et al. [12] examined the spatial variability of REEs in the lakebed sediments with the aim of identifying factors influencing REEs, distinguishing their sources, whether natural or anthropogenic, and assessing the degree of contamination of the sediments. Slukovskii et al. [13] delved into the accumulation of REEs in surface sediments of the Arctic region lakes by analyzing their concentrations and investigating the influence of natural landscapes on REEs within sediments. More recently, Li et al. [14] examined the geochemical characteristics of REEs in subsiding lakebed sediments, exploring the impact of multiple factors on REEs and shedding light on the transport and transformation processes of REEs during subsidence. Thus, it appears that REEs have been widely employed for provenance tracing in a short period of time because their transport and transformation in the environment have little or no fractionation effects, and thus they are uniformly

distributed in particulate matter with essentially no significant changes in their composition and distribution patterns [11,15].

Multi-salt lakes are a distinctive characteristic of this study area, and the study of lakes with different hydrogeological conditions is of great significance. Currently, the study of using REEs to reveal the geochemical processes and environmental evolution of lakes in the region is still in its infancy, especially the provenance tracing. Wang et al. [16] investigated the sediment sequences in Qinghai Lake spanning the past 100 years and analyzed the effects of changes in local land desertification, dust input, and agriculture on sediment deposition. Subsequently, they studied the depositional environment of the sediments in the area using radioactive elements [17]. More recently, Tao et al. [18] conducted an analysis of REEs concentrations in sediments near Qinghai Lake, complemented by petrological and mineralogical methods, to study the provenance in the region.

Previous studies have focused on Qinghai Lake, neglecting investigations of other lakes, which may have led to an incomplete understanding of the geological and climatic evolution of the entire region. For instance, Keluke Lake and Tuosu Lake are hydraulically connected, yet they exhibit markedly distinct lake characteristics, with the former being a freshwater lake and the latter a saline lake. In this study, we expanded the scope and number of sampling efforts beyond Qinghai Lake to include other lakes (Keluke, Tuosu, and Gahai Lake) as well as their respective inflowing rivers. We conducted comparisons across different regions, enhancing our understanding of the geological environment and sediment genesis within the Tibetan Plateau. Additionally, the application of REEs in environmentally sensitive areas has extended their utility in the field of Earth Sciences.

## 2. Study Area and Methodology

### 2.1. Hydrogeological Conditions

This study was conducted in the Northeast Tibetan Plateau (36°14′~37°30′ N, 96°~98° E) (Figure 1a). The average elevation is approximately 4500 m, with higher elevations in the north and lower in the south [19]. Climatically, the study area exhibits the typical plateau continental climate. The average temperature during the warmest month is below 10 °C, and during the coldest month it is below −5 °C [20]. The average annual precipitation ranges between 300 and 400 mm, most of which occurs between June and October [21], and is lower than the annual average evaporation (~927.39 mm) [22].

The main geomorphic features include the northern Zongwulong Mountains, the Buhete Mountain piedmont alluvial inclined plain, the central alluvial plain, and the south alluvial plain [23]. Geologically, the stratigraphy consists mainly of terrestrial and lacustrine deposits of Paleoproterozoic to Neoproterozoic age [24]. The area is primarily composed of Jurassic, Cretaceous, and Tertiary sedimentary rocks, covering basement rocks such as Paleozoic granite and metamorphic rocks. Among them, the Cretaceous sedimentary rocks are predominant and occupy most of the basin's area, while the Cenozoic sedimentary rocks are mainly distributed in the southeastern margin of the basin. In this study, we focus on two regions with different geochemical characteristics.

(1) Region I: Keluke, Tuosu, and Gahai Lakes (Figure 1b).

Keluke Lake (37°14′~37°20′ N, 96°51′~96°57′ E) has an area of 57.9 km$^2$, with an average water depth of 2.94 m. The lake salinity ranges from 0.64 to 0.79 g/L, and it is the largest exhumed freshwater lake [25,26]. The main water sources include Bayin River water (derived from glacial meltwater), seepage groundwater, and natural precipitation [27]. In contrast, Tuosu Lake (37°04′–37°13′ N, 96°50′–97°03′ E) has an area of 165.9 km$^2$ and a maximum depth of 23.6 m. This lake has an average salinity of 27.8 g/L, which is typical of a hydrologically closed saline lake [26,28]. These two lakes share a common supply river, the Bayin River (36°53′~38°11′ N, 96°29′~98°08′ E), which flows from the southeast to the northwest into Keluke Lake (the open lake). The Bayin River is a large permanent river (with a mean discharge of $1.9 \times 10^8$ m$^3$/year). The lake water has a mean residence time of 6 months, based on its total river inflow of $3.41 \times 10^8$ m$^3$ and lake volume of $1.67 \times 10^8$ m$^3$. Therefore, the reason Keluke Lake is freshwater in arid climates is that the main source

of recharge is from snowmelt on the mountains that flow into the river. Salts from this lake are discharged into Tuosu Lake (the endorheic lake) through a connecting freshwater river. The evaporation in Tuosu Lake leads to a gradual accumulation of salts and no discharge, resulting in the formation of the saline lake. The two lakes are dominated by clastic minerals (dolomite and orthoclase), carbonate minerals (calcite, aragonite, dolomite, and hydromagnesite), clay minerals, sulfate minerals, and chlorides. Of these, clastic minerals predominate, followed by carbonate minerals.

Gahai is a typical closed saline lake (37°08′ N, 97°33′ E), with a relatively small area of about 35 km$^2$ and an average water depth of about 8.0 m. There is no perennial surface river inflow, and it mainly relies on atmospheric precipitation and underground submerged recharge.

(2) Region II: Qinghai Lake and nearby rivers in the Qinghai region (Figure 1c).

Qinghai Lake is the largest inland saline lake in China, with an average depth of 18.3 m. The lake is influenced by more than 70 different recharging rivers, with the northwestern part of the lake having more recharging rivers and high runoff, while the southeastern part is relatively sparse and consists mainly of seasonal rivers. Therefore, the main source of recharge for the lake is riverine input. Studies have indicated that the main recharging rivers of Qinghai Lake include the Buha, Shaliu, and Haergai Rivers, which contribute more than 87% of the lake water [29], about 67% of the dissolved burden [30], and sediment discharges from the lake. The water chemistry of the Buha River is influenced by the underlying marine limestones and sandstones, which limit the proportion of carbonate components in the waters of Qinghai Lake [31]. Qinghai Lake is an inland tectonic lake located in the Qinghai Lake fault basin, and thick Quaternary sedimentary strata are developed around the lake basin. A large number of sandstones, shales, and metamorphic rocks are widely distributed on the northern side of the lake; a large number of sandstones, schists, and volcanic rocks are distributed on the western side; and tuffs, metamorphic sandstones, and granites are mainly distributed in the southern part of the Qinghai Nanshan strata. In this study area, Qinghai Lake and the recharging rivers mainly comprise felsic rocks (granite, granodiorite, and felsic volcanic rocks), carbonate rocks, metamorphic rocks (dacite and metavolcanic rocks), and clastic rocks. Of these, the felsic hydrocarbon source rocks have the most significant influence on the chemical composition of the sediments [18].

The Buha River has a total length of 286 km and an area of 14,387 km$^2$, with an average annual runoff of $7.85 \times 10^8$ m$^3$, accounting for 46.9% of the runoff into the lake. The Shaliu River is the second largest recharge river, with a length of 106 km, a basin area of 1500 km$^2$, and an average annual runoff of $2.46 \times 10^8$ m$^3$, accounting for 14.5% of the total runoff. The third largest recharge river is the Haergai River, and the last one is the Daotang River, which is 60 km long, with a watershed area of 727 km$^2$ and a relatively small average annual runoff, and is the only river near Qinghai Lake that flows from southeast to northwest [32]. The Qinghai Gahai (36°57′~37°3′ N; 100°31′~100°36′ E) is a sub-lake of Qinghai Lake and is a closed lake with no river inflow, with a water depth of 8~9.5 m, an area of 47.5 km$^2$, and a salinity of about 31.73 g/L. The lake has a total length of 60 km, a basin area of 727 km$^2$, and an average runoff of small size.

At the time of sampling, there was little difference in water temperatures between the two areas, ranging from 14.0 to 17.0 °C and averaging 15.8 °C. There was no rain when we sampled, and the river samples did not contain samples taken during heavy rain. River inputs were present in both regions, and the rivers did not dry up.

## 2.2. Sampling Design and REEs Analysis

In this study, 29 samples were collected during the summer of 2018, covering two regional lakes in the Northeast Tibetan Plateau as well as nearby rivers (Table S1). In summer, Region I Keluke Lake is classified as an open lake, and it receives glacial meltwater from the Tian Shan Mountains through the Bayin River, which subsequently flows into Tuosu Lake. This lake exhibits hydraulic connectivity, and the river inputs disrupt the distribution of the particulate matter within the lake, causing a significant portion to remain

in a suspended state. In Region II, the flux of debris carried by rivers near Qinghai Lake in summer is high, and the contribution of the Buha and Shaliu rivers in the north can reach 90% [32]. Riverine inputs are present in both regions, with the two lakes in Region I being connected by a river with a hydraulic link and Qinghai Lake in Region II being recharged by nearby rivers. The retention time of particulate matter in the water body during the descent of detritus from the water body to the lake surface sediments is variable. Then, for the particulate matter to undergo sub-level changes in the water body, it may be influenced by the water body's organic matter and sub-level water-rock interaction processes such as dispersion, coagulation, and re-dispersion of particulate matter. Therefore, the rare earth distribution pattern and content can provide relevant information. In order to determine the degree of influence of transferring suspended particulate matter on the partitioning of rare earth elements due to biological involvement in the process of suspended matter from river water to sediments in both regions, sampling locations were targeted in this study, with samples collected mainly at the river equidistant point and at the point where the river enters the lake. All samples were collected using high-density polyethylene bottles (HDPE, Nalgene 2002-0032) and processed by water filtration through a Nalgene manual vacuum filtration system (Nalgene 300-4100; 6133-0010) and a 0.8 μm hybrid cellulose filter membrane (MilliPore AAWP04700). Subsequently, the filtered particulate samples were determined for trace elements and REEs.

All particle-filtered samples were weighed for trace element analysis by using inductively coupled plasma mass spectrometry (ICP-MS, Agilent 7700×, Santa Clara, CA, USA). The processing and analytical procedures for trace elements were as described in Ma et al. [33]. The samples were first heat-dissolved (3–7 days) with $HNO_3$-$H_2O_2$, HF-$HNO_3$-$HClO_4$, 7N $HNO_3$, and 6N HCl at 130 °C until the samples were completely dissolved; the dissolved samples were quantitatively portioned for trace element analysis. The digested solution aliquot was cooled and diluted to 5 mL with $HNO_3$ (1% *v/v*) in a fume hood. All the chemicals used were of analytical grade (Beijing Chemical Reagent Research Institute). Quality assurance was provided by analyzing certified reference material (Calstd, including REE, EN, Multi-EN, and Ti). A duplicate sample and a blank sample, along with each batch of nine samples, were analyzed. Replicate analysis of these reference materials revealed a high level of accuracy; the precision for most trace element measurements is estimated to be 5% according to the analysis of the Calstd standards. All samples were treated and measured in the Key Laboratory of Western China's Environmental Systems, Lanzhou University.

The study used an inductively coupled plasma mass spectrometer (ICP-MS) to complete the process, which included sample pre-treatment, separation, and determination of REEs. The particulate samples were first dried and milled to remove lake water and dirt and to ensure homogeneity of the samples. Next, the dried particulate samples were dissolved, ablated, and treated using a mixture of acids (nitric acid: hydrofluoric acid = 3:1), the treated samples were in a clear state, and then cooled and the samples were diluted using distilled water. Meanwhile, a series of standard solutions of REEs with different concentrations were prepared for the creation of standard curves, and the parameters of ICP-MS were set and optimized. Then, the ion exchange resin column was used to separate the REEs in the samples, followed by the elution of the resin using hydrochloric acid, and the elution solution was fed into the ICP-MS for determination together with the calibration standard. In this study, Rh was used as the internal standard, and to ensure the accuracy of the experiment, the sample recovery was kept above 95%, and ten samples were inserted into parallel samples, standard samples, and blanks for each test. The precision and accuracy of the REEs tests were controlled within ±5%. The methods adopted herein are elaborated upon by Cheng et al. [34] and Shen et al. [35].

### 2.3. Statistical Analysis

Survey data were organized using Excel 2021 and expressed as "mean ± standard deviation". Statistical analyses were performed using GraphPad Prism 9. The statistical differences in the total suspended matter (TSM) and REEs concentrations between particulate matter in lakes and rivers are compared using an unpaired *t*-test. All data were tested for normality (Shapiro-Wilk test) and homogeneity of variance (F test for *t*-test) before the test of significance. The *p* value (<0.05) was considered significant.

## 3. Results

In this study, we primarily analyzed the distribution characteristics of REEs in two regional lakes and their respective inflowing rivers. Subsequently, we examined the differences in REEs between these two regions and employed various methods to trace the sources of particulate matter in the area.

### 3.1. Particulate Matter Concentrations and Chemical Components in Two Regions

The lakes in Region I have neutral or weakly alkaline waters. The pH range of Keluke Lake is 7.52–7.77, which is lower than that of Tuosu Lake (8.83–8.88), and Gahai has a pH of 7.84. The cations of these lakes are dominated by sodium ions, and the anions are dominated by chloride ions. The concentration of an ion (or ion pair) remained essentially the same throughout the same water body. In addition, the pH of Qinghai Lake and its sub-lakes in Region II was 8.76–8.89, which was higher than that of the recharge rivers (the pH range of the three rivers was 7.53–7.94). The pH of Xiligou Lake was also weakly alkaline, ranging from 7.53–8.51. Qinghai Lake was dominated by sodium and chloride ions, whereas the recharging rivers were dominated by calcium and sulfate ions [36].

Concentrations of TSM in Keluke Lake ranged from 95.56–7600.00 μg/L and were significantly lower than in Tuosu Lake (3849.05–11161.29 μg/L, *t* = 2.880, *p* = 0.0164) (Table S2). The trace element composition is divided into three categories. (1) The first category consists of elements that reflect the source of the particulate debris, including thorium (Th) and aluminum (Al). (2) The second category consists of six elements that reflect the degree of weathering of the particulate matter, namely, lithium (Li), magnesium (Mg), titanium (Ti), strontium (Sr), molybdenum (Mo), and uranium (U). (3) The third category consists of four elements that reflect biological productivity, including iron (Fe), zinc (Zn), cadmium (Cd), and barium (Ba). In Region I, the upper continental crust is primarily composed of granite, and its chemical composition is dominated by oxygen, silicon, and aluminum. Granite is a type of intrusive rock formed during the solidification of magma beneath the Earth's surface, and its main minerals include quartz, potassium feldspar, and acid plagioclase. Furthermore, uranium and thorium are concentrated in the uppermost layer of the Earth's crust, with concentrations approximately four times higher than those of basalt and about thirty times higher than those of the mantle. The Al/Th values are less variable, while the Th/U values fluctuate slightly but remain below the average abundance of uranium and thorium in the Earth's crust, with a difference of no more than an order of magnitude (Table S2). This indicates that the source of the particulate matter in the two lakes is relatively stable, and the particulate matter in both lakes has a similar origin.

In Region II, the total suspended matter (TSM) concentrations in Qinghai Lake and its neighboring rivers varied between 6012 and 415,960 μg/L, which was significantly higher than that in Region I (Table S3, *t* = 4.996, *p* < 0.0001). This difference suggests significant differences in TSM concentrations between geographic regions. The Al/Th values of Qinghai Lake and its supplying rivers were in the same order of magnitude, indicating that the source of particulate matter was relatively stable, mainly due to riverine inputs of detrital material. In addition, it was found that the concentrations of type III elements in Qinghai Lake and its sub-lake (Qinghai-Gahai) were lower than those in the supplying rivers, which may be due to the significant differences between the lake ecosystem and the riverine environment. The rich biological communities within the lake could absorb

and utilize dissolved nutrients in the water, resulting in more elements being converted from the particulate matter to the dissolved state, which reduced the concentrations of elements in the particulate matter. However, no significant difference was observed in this study between Qinghai Lake and its supplying rivers in terms of easily transportable elements ($t = 1.716$, $p = 0.0922$). This may be related to physicochemical processes within the lake and mixing with the river, but it also indicates that Qinghai Lake is influenced by the supplying river and that elemental concentrations between the two are within the same order of magnitude.

### 3.2. Region I: Distribution Characteristics of REEs

The concentration variations of particulate REEs differed in lakes in Region I. In Keluke Lake, REEs ranged from 0.04 to 70.16 ppm, while in Tuosu and Gahai Lakes, REEs ranged from 0.03 to 9.22 and 4.46 to 30.87 ppm, respectively (Table S4). The spatial variability coefficients (CV) for Keluke Lake ($CV_{KLK} = 1.43$) were greater than those in Tuosu ($CV_{TS} = 1.00$) and Gahai Lake ($CV_{GH} = 0.76$), indicating that the spatial distribution of REEs within the three lakes was heterogeneous.

The average concentration of $\Sigma$REE in Keluke Lake was 138.89 ppm, which was significantly higher than that in Tuosu Lake ($\Sigma REE_{TS} = 22.78$ ppm) ($t = 1.733$, $p = 0.0004$) (Table 1). However, the REEs of all three lakes exhibited enrichment of light rare earths (LREE) and depletion of heavy rare earths (HREE) (Figure 2). The mean $\Sigma$LREE/$\Sigma$HREE ratios in Keluke, Tuosu, and Gahai lakes were 3.07, 2.46, and 2.43, respectively, indicating that LREE was more enriched than HREE in the particulate matter of the three lakes and that LREE was the dominant contributor to $\Sigma$REE, but with different degrees of enrichment.

**Table 1.** Region I: Parameter indexes of rare earth elements in lake particulate matter.

| Site | Station | $\Sigma$REE | $\Sigma$LREE/$\Sigma$HREE |
|---|---|---|---|
| | KLK1-M | 47.79 | 2.67 |
| | KLK1-B | 12.60 | 3.97 |
| | KLK2-T | 22.08 | 3.57 |
| Keluke Lake | KLK2-M | 28.62 | 3.76 |
| | KLK2-B | 319.71 | 2.73 |
| | KLK3-T | 350.42 | 2.77 |
| | KLK3-M | 191.01 | 2.01 |
| | TS1-T | 19.61 | 1.93 |
| | TS1-M | 45.85 | 2.56 |
| Tuosu Lake | TS2-T | 10.27 | 2.50 |
| | TS2-M | 16.33 | 3.08 |
| | TS3 | 21.84 | 2.25 |
| Gahai Lake | GH1 | 158.14 | 2.43 |

Note: The letters S, M, and B in the station name represent samples from the surface, middle, and bottom of the lake, respectively.

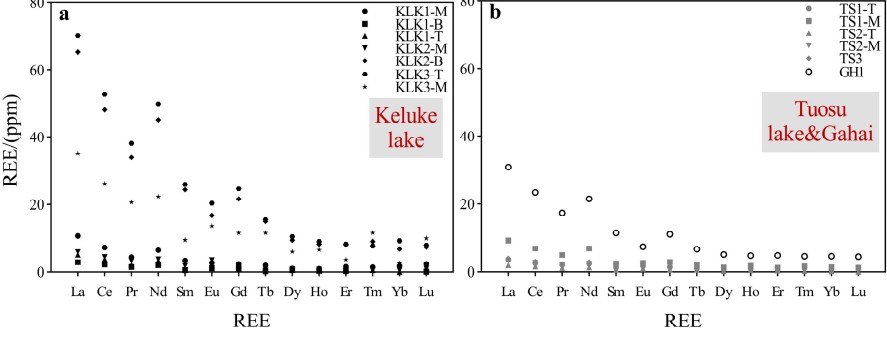

**Figure 2.** Region I: Concentrations of REEs in particulate matter from three lakes. (**a**) Keluke Lake. (**b**) Tuosu Lake and Gahai.

Furthermore, the study observed irregular fluctuations in ΣREE concentrations with depth in Keluke and Tuosu Lakes (Table 1), which may be attributed to the current impact perturbation of the Bayin River into the lake and the wind-driven water circulation perturbation of particulate matter [37]. The concentration of ΣREE in KLK2 increased with depth, while KLK1 and KLK3 showed the opposite trend. The concentration of ΣREE in TS1 and TS2 also increased with depth. The particulate matter in these lakes primarily existed in a suspended state, and the input from the Bayin River might disturb the distribution of particulate matter, potentially leading to vertical variations in ΣREE concentrations [38]. In addition, the residence time in Keluke Lake was less than 10 months, while the residence time in Tuosu Lake was approximately 1 year, indicating that particulate matter within both lakes tended to be in suspension during the survey period. Another possibility is that the Tuosu Lake contains organic matter from lower algal plants; biological activities could affect the distribution of REEs [39]. The presence of abandoned mines around Tuosu Lake could be another potential factor [40].

### 3.3. Region II: Distribution Characteristics of REEs

Region II covers Qinghai Lake and its surrounding rivers, where concentrations of REEs showed significant differences. Specifically, the mean ΣREE concentrations in Qinghai Lake and its subsidiary lake (QHGH) were 16.93 and 0.68 ppm (Table S5), respectively, both significantly lower than the concentration levels in the recharging rivers ($t = 2.236$, $p = 0.0005$). REEs typically exhibit high stability in terrestrial surface environments, but they are primarily transported in the form of particulate matter in rivers [41]. The high concentrations of REEs in these particles constitute the primary form of REEs presence in river water, primarily originating from the chemical weathering processes of rocks within the watershed [18,42].

The average ΣREE concentrations in these three rivers also exhibit significant differences. Specifically, the Shaliu River has an average ΣREE concentration of 174.23 ppm, which is significantly higher than that of the Buha River ($ΣREE_{BH} = 29.54$ ppm) and the Haergai River ($ΣREE_{HRG} = 19.79$ ppm) ($t = 6.743$, $p = 0.0025$). This discrepancy may be associated with the hydrological characteristics, water chemistry composition, as well as particle types and concentrations within these rivers. According to Tao et al. [18], the sediments of the Buha River are rich in polycrystalline quartz, while the other two rivers are rich in a large amount of granite fragments and a small amount of metamorphic rocks, and the type of sediments in different rivers affects the differences in REEs. Furthermore, the concentration of REEs in the Shaliu River and the Buha River varied by a larger magnitude because the amount of suspended sediment in the rivers also significantly affected the concentration of REEs [43]. However, both lakes and rivers exhibit enrichment of LREE and depletion of HREE, albeit to varying degrees (Figure 3). The high ΣLREE/ΣHREE ratio further confirms the predominance of LREEs over HREEs (Table 2).

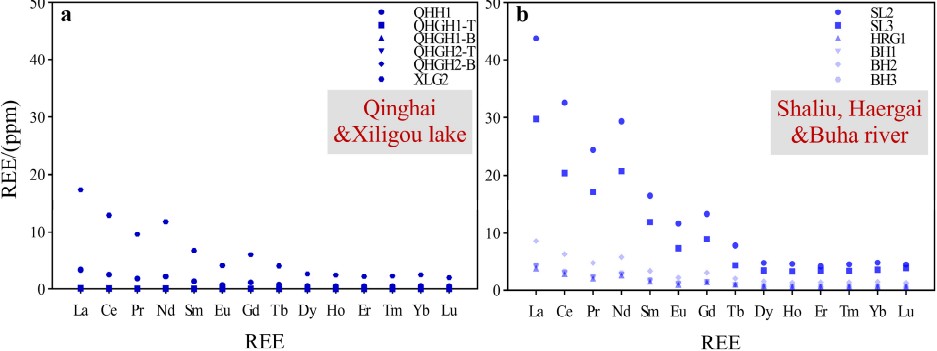

**Figure 3.** Region II: Concentrations of REEs in particulate matter from lakes and rivers. (**a**) Qinghai Lake and its subsidiary lakes (Qinghaigahai and Xiligou Lake). (**b**) the Shaliu, Haergai, and Buha rivers.

**Table 2.** Region II: Parameter indexes of rare earth elements in lake and river particulate matter (N is chondrite standardization).

| Site | Station | ΣREE | ΣLREE/ΣHREE |
|---|---|---|---|
| Shaliu River | SL2 | 206.76 | 3.27 |
| | SL3 | 141.69 | 3.12 |
| Haergai River | HRG1 | 19.79 | 2.27 |
| | BH1 | 21.31 | 2.39 |
| Buha River | BH2 | 44.81 | 2.29 |
| | BH3 | 22.50 | 2.72 |
| Qinghai Lake | QHH1 | 16.93 | 2.50 |
| | QHGH1-T | 0.66 | 2.39 |
| Qinghaigahai Lake | QHGH1-B | 0.51 | 2.39 |
| | QHGH2-T | 0.68 | 2.69 |
| | QHGH2-B | 0.85 | 2.30 |
| Xiligou Lake | XLG2 | 86.74 | 2.57 |

*3.4. Region I vs. Region II*

There is no significant difference in ΣREE between two regions ($t = 1.240$, $p = 0.2275$), which may be attributed to regional hydrological and sedimentary conditions. However, both are characterized by relatively enriched LREE and lower HREE. LREE is primarily adsorbed to particulate matter by organic materials and detrital substances, whereas HREE tends to form stable complexes and remain in the lake water [44]. The degree of enrichment differs between the two regions.

Open Keluke Lake in Region I is more enriched in LREE than Tuosu Lake and Region II, possibly related to lake properties, river-lake interactions, and terrestrial inputs. For instance, the lower salinity and dissolved matter of freshwater lakes make it easier for LREE to accumulate on particle surfaces, whereas in saline lakes, enriched in salts and dissolved substances, the degree of LREE enrichment is lower [45].

## 4. Discussion

*4.1. Chondrite Standardized Partitioning Patterns*

The primary focus of source tracing using the partitioning patterns of REEs is on the geometry of the ligand curves rather than the absolute abundance of the elements [46]. The data presented in this study are normalized using Boynton chondrite values [47].

In Region I, the particulate matter from Keluke Lake exhibited greater differentiation between light and heavy rare earths, with mean values of $29.67 \pm 48.81$, $10.40 \pm 18.46$, and $3.63 \pm 0.92$ for $La_N/Yb_N$, $Gd_N/Yb_N$, and $La_N/Sm_N$, respectively (Table 3, Figure 4a). In contrast, at Tuosu Lake, the mean values of these ratios were $7.62 \pm 1.67$, $2.54 \pm 0.66$, and $3.18 \pm 0.53$, respectively, indicating a lower degree of differentiation between light and heavy rare earths than at Keluke Lake (Table 3 and Figure 4b).

**Table 3.** Region I: Parameter indexes of rare earth elements in lake particulate matter (N is chondrite standardization).

| Site | Station | $La_N/Yb_N$ | $Gd_N/Yb_N$ | $La_N/Sm_N$ |
|---|---|---|---|---|
| Keluke Lake | KLK1-M | 6.19 | 1.35 | 3.19 |
| | KLK1-B | - | - | 5.06 |
| | KLK2-T | 129.14 | 48.01 | 4.67 |
| | KLK2-M | 11.57 | 2.59 | 3.39 |
| | KLK2-B | 9.53 | 3.16 | 2.67 |
| | KLK3-T | 7.58 | 2.67 | 2.69 |
| | KLK3-M | 14.03 | 4.60 | 3.71 |
| Tuosu Lake | TS1-T | 6.97 | 2.57 | 2.95 |
| | TS1-M | 7.10 | 2.18 | 3.86 |
| | TS2-T | 10.54 | 3.55 | 2.67 |
| | TS2-M | 6.30 | 1.76 | 3.61 |
| | TS3 | 7.21 | 2.63 | 2.79 |
| Gahai Lake | GH1 | 6.74 | 2.41 | 2.70 |

Note: "-" represents "no data".

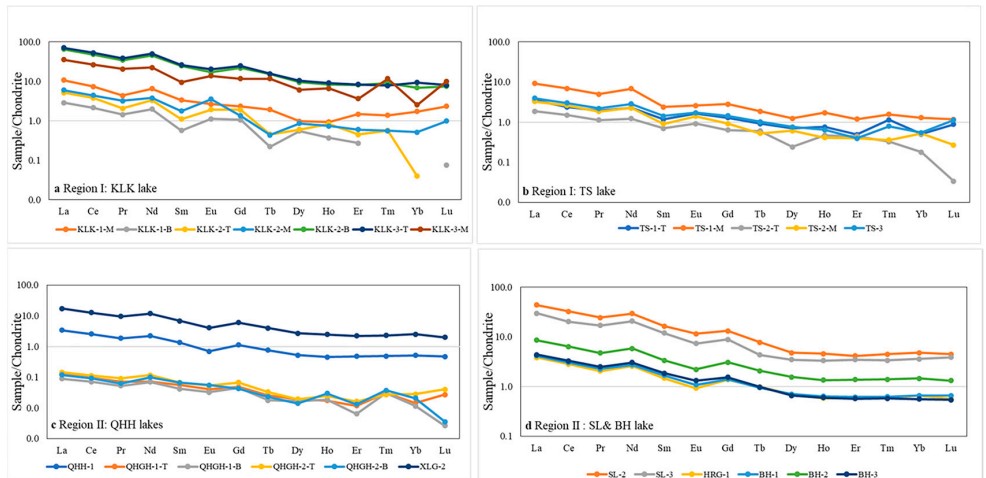

**Figure 4.** Standardized distribution pattern of REEs globular meteorites in lakes and rivers in two regions. (**a**) Region I: Keluke Lake. (**b**) Region I: Tuosu Lake. (**c**) Region II: Qinghai Lake and its subsidiary lakes (Qinghaigahai and Xiligou Lake). (**d**) Region II: Shaliu and Buha rivers.

In Region II, particulate matter from Qinghai Lake and its recharging rivers demonstrated a relatively low degree of rare-earth fractionation, with $La_N/Yb_N$, $Gd_N/Yb_N$, and $La_N/Sm_N$ of $7.01 \pm 1.29$, $2.54 \pm 0.56$, and $2.38 \pm 0.25$, respectively (Table 4 and Figure 4c,d). Compared to the Upper Continental Crust (UCC, $La_N/Yb_N$ of 10.5 [18]), the REEs from Keluke Lake showed a higher degree of fractionation, whereas the samples from Tuosu Lake and Region II showed a relatively lower pattern of rare earth element fractionation.

**Table 4.** Region II: Parameter indexes of rare earth elements in lake and river particulate matter (N is chondrite standardization).

| Site | Station | $La_N/Yb_N$ | $Gd_N/Yb_N$ | $La_N/Sm_N$ |
|---|---|---|---|---|
| Shaliu River | SL2 | 9.11 | 2.75 | 2.66 |
| | SL3 | 8.27 | 2.48 | 2.50 |
| Haergai River | HRG1 | 5.83 | 2.09 | 2.61 |
| Buha River | BH1 | 6.34 | 2.15 | 2.47 |
| | BH2 | 5.89 | 2.11 | 2.54 |
| | BH3 | 7.82 | 2.74 | 2.35 |
| Qinghai Lake | QHH1 | 6.55 | 2.18 | 2.53 |
| Qinghaigahai Lake | QHGH1-T | 8.66 | 3.27 | 2.24 |
| | QHGH1-B | 7.84 | 3.89 | 2.18 |
| | QHGH2-T | 5.76 | 2.05 | 1.81 |
| | QHGH2-B | 5.15 | 2.38 | 2.16 |
| Xiligou Lake | XLG2 | 6.91 | 2.42 | 2.55 |

The REEs partitioning patterns in two regions are closely similar, and both are of the gentle right-handed tilt type, characterized by a steep LREE distribution and a relatively flat HREE distribution, indicating that the particulate matter has the same material sources and formation process. In Region I, Keluke Lake and Tuosu Lake are connected by the Bayin River. This suggests that the provenance for these two lakes is consistent, primarily originating from exposed granite rocks, which were weathered and transported by rivers before being deposited into the lake system. The Qinghai Lake and river particulate matters in Region II consist mainly of lacustrine rocks (granites, granodiorites, and lacustrine volcanics), carbonates, and clastic rocks [18], which have the greatest influence on particulate matter chemistry; that is, they have undergone a certain degree of chemical weathering from recharging rivers to the lake and have been affected by some recycled clastic materials. The particulate matter data in this study also support the rare earth element partitioning pattern of Qinghai Lake sediments.

The difference lies in the Eu anomalies. Studies have revealed that weak redox environments may cause Eu-containing minerals to dissolve initially, resulting in the leaching of a portion of the Eu from particulate matter [48]. Eu is more sensitive to the effects of the redox environment relative to other rare earth elements, and it has the ability to form a different $Eu^{2+}$ morphology than the other trivalent REEs, which causes it to show a distinct concave-convex feature in the normalized model (Figure 4). It is important to point out that the reduction of trivalent Eu to divalent Eu is relatively difficult under ambient temperature and pressure conditions [49], and this conversion can only be achieved in an environment with a redox potential less than −350 Mv [50]. However, based on the sampling depth and previous findings, the environments in both regions possess sufficient dissolved oxygen with redox potentials greater than 0 [26,32]. Therefore, the contribution of the redox environment to the Eu anomalies may be weak. In Region I, the positive Eu anomalies may originate from the dissolution of Eu-rich minerals (clastic and carbonate minerals). The existence of an open lake system like Keluke Lake, which receives glacial meltwater from the Tien Shan region, can introduce Eu-rich minerals into the Bayin River, and the dissolution of these minerals will elevate the Eu concentration as they dissolve. In Region II, the negative Eu anomaly is consistent with previous studies [18], which may be related to the low Eu-rich host rocks, and the bedrock of Qinghai Lake and the recharging rivers (mainly long quartzite) has relatively low Eu concentrations. Therefore, the anomalies in Eu can be attributed to the dissolution of Eu-rich minerals. In Region I, the positive anomalies of Eu originate from the dissolution of Eu-rich minerals, while in Region II, the negative anomalies of Eu are inherited from the host rock.

### 4.2. Parameter Ratio of REEs

The REEs composition and partitioning characteristics of lake particulate matter can be used for source discrimination, which can be characterized using the characteristic parameters of REEs [51]. Among the previously mentioned, the ΣLREE/ΣHREE ratio can be used to reflect the source of particulate matter. In addition, δCe/∑REE and δCe/δEu are also of great significance. Diagenesis can affect Ce anomalies in particulate matter, resulting in good correlations between δCe and ∑REE and δEu [52,53]. The results of the study revealed that there is no significant correlation between these elements in both regions with *p*-values significantly greater than 0.05 (Figure 5), indicating that diagenesis does not have a significant effect on the REEs of the particulate matter and that the composition of the particulate matter is mainly controlled by the type of the host rock (Region I: clastic and carbonate minerals; Region II: the felsic hydrocarbon source rocks), and therefore, the source of the particulate matter can be further analyzed by using the discriminant function of the REEs.

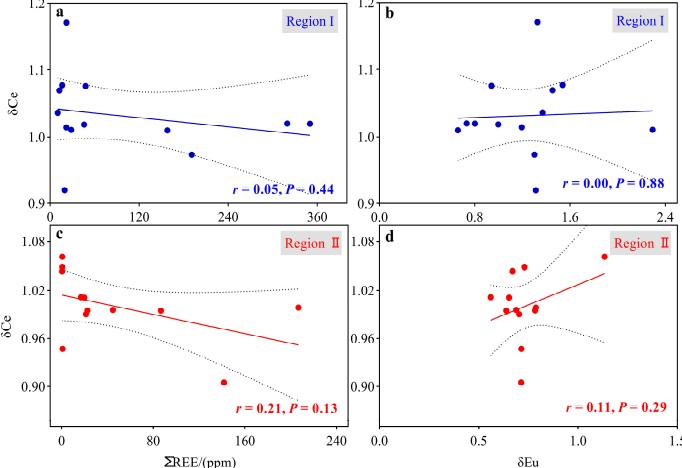

**Figure 5.** Correlation between δCe and ∑REE, δCe and δEu of particulate matter in Region I (**a**,**b**) and Region II (**c**,**d**).

### 4.3. Discriminant Function in Region II

According to the standard partitioning model and characteristic parameter analysis, the particulate matter of Qinghai Lake in Region II mainly originates from terrestrial source inputs, of which the terrestrial source debris is mainly transported to the lake through rivers. In order to more accurately determine the degree of contribution of each recharging river to the particulate matter of Qinghai Lake, the study utilized the discriminant function (DF) to reveal the degree of similarity between the particulate matter of Qinghai Lake and that of the land-source material [54].

$$\mathrm{DF} = \left| \frac{\frac{C_{1X}}{C_{2X}}}{\frac{C_{1L}}{C_{2L}}} - 1 \right| \tag{1}$$

where "$C_{1X}/C_{2X}$" and "$C_{1L}/C_{2L}$" are the ratios of two relatively stable light rare earth elements ($C_{Sm}/C_{Nd}$) in particulate matter and terrestrial source material, respectively. The recharge rivers, as potential source areas of Qinghai Lake, showed significant differences in their DF values, i.e., $DF_{BH} < DF_{HRG} < DF_{SL}$ (Figure 6). On the contrary, the smaller DF values indicated that the particulate matter samples were close to the recharged land-based sources. Thus, the Buha River contributed the most to the particulate matter in Qinghai Lake, followed by the Haergai and Shaliu Rivers. It was found that the degree of contribution fit well with the runoff of these rivers, and the Buha River, which had the highest runoff, contributed the most to the particulate matter in Qinghai Lake. Our results are consistent with the research of Cui and Li [55].

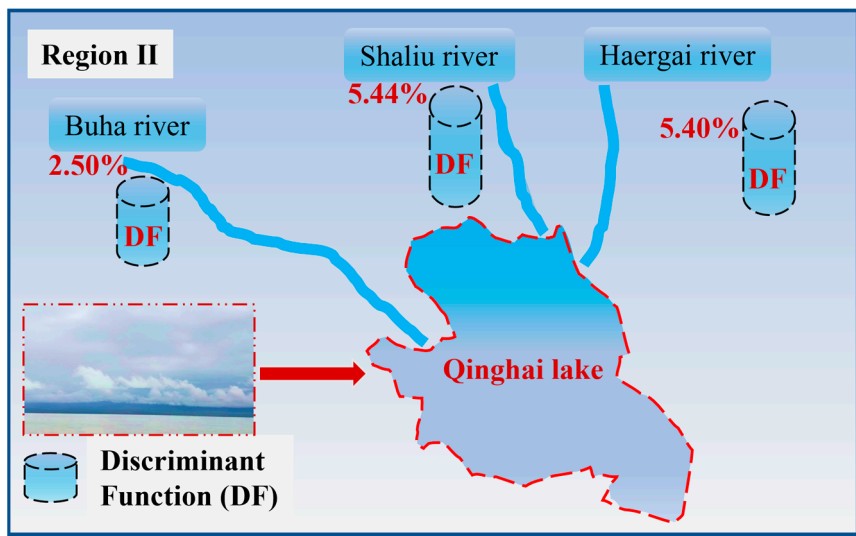

**Figure 6.** The function of discrimination function (DF) values for particulate matter in Region II.

### 4.4. Indication of REEs

The results from the two regions demonstrated consistent normalization patterns, similar REEs sources, and common geochemical characteristics of the particulate samples. In addition, the LREE enrichment, relative HREE depletion, and Eu anomalies all indicated that the REEs were migrated and transported as a whole during chemical weathering and deposition, which is consistent with the findings of Liu et al. [56]. The present study demonstrates that studying the provenance of REEs in lake particles is essential to studying elemental geochemical behavior, especially in environmentally sensitive areas. It not only reveals the geochemical characteristics of REEs, but also reflects the processes of surface weathering and water evolution. This study mainly provides preliminary insight. Solid fugacity (solid speciation) studies of REEs in particles in future work could contribute to a comprehensive understanding of REEs geochemical processes, such as the proportional distribution of REEs in the dissolved phase of the water column and in the solid particulate

phase. Furthermore, in addition to natural processes, human activities (e.g., abandoned uranium mines near Tuosu Lake and agricultural activities [57], etc.) can interfere with the redistribution of REEs in rivers, lakes, and soils. Therefore, REEs in lake particulate matter in sensitive areas record important information on sedimentary geology, environmental changes, and human activities and should be utilized in various applications such as geological resource exploration, pollution and environmental quality monitoring, and ecosystem health assessment.

**5. Conclusions**

In this study, we investigated two regions in the Northeast Tibetan Plateau, namely Region I (Keluke, Tuosu, and Gahai Lakes) and Region II (Qinghai Lake and nearby rivers). The characteristics and differences of REEs in the particulate matter of the two regions were first analyzed, followed by the determination of regional provenances, and finally the importance of REEs in provenance studies was analyzed. The main conclusions are as follows:

1.  The differences in REEs concentrations were relatively large in both regions, and the spatial distribution was heterogeneous. The ΣREE concentrations of lake particulate matter in Region I varied irregularly in depth, while the differences in ΣREE concentrations in Region II were correlated to the type and amount of particulate matter.
2.  There is no significant difference in ΣREE between the two regions, which are both characterized by relatively enriched LREE and lower HREE, but the degree of enrichment differs between the two regions.
3.  REEs partitioning is similar in the two regions, indicating that the source and formation of the particulate matter are consistent. Particulate matter in Region I predominantly originates from granite rocks undergoing weathering and transportation through rivers; the positive anomalies of Eu originate from the dissolution of Eu-rich minerals. Region II particulate matter is affected by chemical weathering and partial recycling of detrital material; the negative anomalies of Eu are inherited from the host rock. Diagenesis had no significant effect on the particulate rare earth elements, and different rivers contributed differently to Qinghai Lake.
4.  This study primarily provides a preliminary understanding of REEs in lake particles, assessing particle changes during the water-to-sediment process and their provenance indication. The solid speciation of REEs in particles is also an important part of future studies of the geochemical behavior of REEs, which is crucial for provenance studies.

**Supplementary Materials:** The following supporting information can be downloaded at: https://www.mdpi.com/article/10.3390/w15203700/s1, Table S1: Basic information of sample collection in lakes and rivers; Table S2: Region I: The element concentration (pg/g) of the first type (elements related to the source of particulate detritus) and the third type (elements related to biological productivity); Table S3: Region II: The element concentration (pg/g) of the first type (elements related to the source of particulate detritus) and the third type (elements related to biological productivity); Table S4: Region I: Composition characteristics of rare earth elements in lake particulate matter (ppm); Table S5: Region II: Composition characteristics of rare earth elements in lake and nearby river particulate matter (ppm).

**Author Contributions:** Conceptualization, P.Z.; methodology, P.Z. and Z.Z.; investigation, L.L. (Lihua Liang), L.L. (Lei Li) and C.C.; data curation, P.Z., Z.Z. and C.C.; writing—original draft preparation, P.Z. and Z.Z.; writing—review and editing, P.Z. and Z.Z.; supervision, P.Z. and R.L.E.; funding acquisition, P.Z.; guidance, R.L.E. All authors have read and agreed to the published version of the manuscript.

**Funding:** This research was funded by the National Natural Science Foundation of China (No. 42173027; 41873013; No. 41888101), Everest Talent Plan Project (Grant No. 10912-KYQD2022-09482), the Scientific Research Start-up Funds of Xi'an Jiaotong University (No. xxj032019007), the 111 Project of China (No. D19002), and the U.S. NSF (No. 1702816).

**Data Availability Statement:** Not applicable.

**Acknowledgments:** This study was supported by the National Natural Science Foundation of China (No. 42173027), the National Natural Science Foundation of China (No. 41873013; No. 41888101), Everest Talent Plan Project (Grant No. 10912-KYQD2022-09482), the Scientific Research Start-up Funds of Xi'an Jiaotong University (No. xxj032019007), the 111 Project of China (No. D19002), and the U.S. NSF (No. 1702816).

**Conflicts of Interest:** The authors declare no conflict of interest.

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
