# Peer review of "Provenance Indication of Rare Earth Elements in Lake Particulates from Environmentally Sensitive Regions"

_water, doi:10.3390/w15203700_

Round 1
Reviewer 1 Report
Paper deals with a somewhat rare subject (rare earth elements) and a still poor geographic area (Tibetan Plateau). Authors did a very nice job based on adequate methods and reaching some very interesting conclusions. All data collected was used during the discussion and a profuse and recent literature was collected to support the paper conclusions. I am very pleased to recomment publication of the article as it is.-
Author Response
Dear Editor-in-Chief,
We would like to thank you for giving us a chance to revise and resubmit the manuscript (“Provenance indication of rare earth elements in lake particulates from environmentally sensitive regions”, water-2638831). We also thank the two reviewers for their constructive suggestions, which helped us improve the quality of the manuscript.
We have modified the manuscript according to the comments and have carefully polished the language. The revised portions are marked in blue.
The main corrections and the point-to-point response to the Reviewer 1 are provided below:
Response to Reviewer 1
Paper deals with a somewhat rare subject (rare earth elements) and a still poor geographic area (Tibetan Plateau). Authors did a very nice job based on adequate methods and reaching some very interesting conclusions. All data collected was used during the discussion and a profuse and recent literature was collected to support the paper conclusions. I am very pleased to recomment publication of the article as it is.-
Response:
Thanks very much for taking your time to review this manuscript. We really appreciate the thoughtful and positive comments you have provided.
Acknowledgements
This MS have all been modified according to the constructive suggestions from the Editor and two Reviewers. Sincerely thanks again for your constructive comments! Best wishes to Water.

Reviewer 2 Report
In the manuscript “Provenance indication of rare earth elements in lake particulates from environmentally sensitive regions”. Authors used REEs because have been widely employed for provenance tracing. In this study, they analyzed the distribution characteristics of REEs in two regional lakes and their respective inflowing rivers. Subsequently, they examined the differences in REEs between these two regions and employed various methods to trace the sources of particulate matter in the area. The manuscript can improve by adressing the next issues:
They only collected and studied 29 samples, covering two regional lakes in the Northeast Tibetan Plateau as well as nearby rivers. Explain the selected points. It seems few samples to made a good statistics analyses, explain and support the selection.
Methods.
Explain clearly this methodology employed or put the references used “the ion exchange resin column was used to separate the REEs in the samples, followed by the elution of the resin using hydrochloric acid and the elution solution was fed into the ICP-MS.
I do not understand the used of an ion column in the methodology because ICP-Ms is a multielement analyses, and references indiacted the ICP-Ms determination.
Explain clearly and discuss the differences between top, middle and bottom samples, there are few samples to concluded, and is not clear.
Therefore, the anomalies of Eu can be attributed to the dissolution of Eu-rich minerals. In Region I, the positive anomalies of Eu originate from the dissolution of Eu-rich minerals, while in Region II, the negative anomalies of Eu are inherited from the host rock, is not clear with the results so please support these conclusion.
The host rock geochemistry is not presented and discuss in the manuscript.
In the manuscript is not clear The solid speciation of REEs in particles so the conclusion is not support by the results, please clarify
Round 2
Reviewer 2 Report
The authors answered and resolved the points raised in review round 1. The manuscript is suitable for publication.